# Work for self or others? Two different kinds of burnout in China

Mengjiao Yin*, Yingying Xia

School of Business, Wuxi Taihu University, Wuxi, Jiangsu, China

* yinmj@wxu.edu.cn

## Abstract

Although much research has been done on occupational burnout, most studies focus on employees (EM), with insufficient attention given to the self-employed (SE). This study adopts a novel quantitative approach using methods like the entropy weight method, multiple linear regression, and the Boruta algorithm to analyze burnout differences between EM and SE. Results indicate that EM are more influenced by stress and external, deficiency-induced burnout, while SE are driven by interest and internal, imbalance-induced burnout. Social network support is a key factor in predicting burnout for both groups. Interestingly, increased financial support from family raises burnout risk for EM, while greater social network support increases burnout risk for SE. These findings reflect inherent differences in work patterns and resource mobilization strategies. This research expands the application of the Job Demands-Resources model and provides insights for tailored mental health interventions across different occupational contexts.

## 1 Introduction

According to the World Health Organization, occupational burnout is a syndrome conceptualized as resulting from chronic workplace stress that has not been successfully managed. It is characterized by three dimensions: 1) feelings of energy depletion or exhaustion; 2) increased mental distance from one's job, or feelings of negativism or cynicism related to one's job; and 3) reduced professional efficacy [1].

Occupational burnout is a prevalent phenomenon in professional environments worldwide. According to a Gallup report titled "Employee Burnout: Causes and Cures," 76% of employees experience burnout at least occasionally, with 28% reporting they feel "very often" or "always" burned out at work [2]. In the United States, 77% of professionals have experienced burnout in their current job, more than half of whom have felt it repeatedly. The European Working Conditions Telephone Survey 2021 found that 17% of participants reported experiencing "emotional exhaustion" or "burnout" [3]. In EU countries, the prevalence of occupational burnout ranges from 4.3% in Finland to 20.6% in Slovenia; in non-EU countries, it varies from 13% in

**Data availability statement:** https://www.kaggle.com/datasets/jojoyin/clds-2018.

**Funding:** This study was supported by funding from the following projects: Philosophy and Social Sciences Research Project of Jiangsu Higher Education Institutions (Grant No. 2025SJYB0693 to MY), Qinglan Project of Jiangsu Province, Teaching Reform Program (Grant No. 25JGYJ30 to MY), and National Natural Science Foundation of China (Grant No. 72573117 to MY). The funders had no role in study design, data collection and analysis, decision to publish, or preparation of the manuscript.

**Competing interests:** NO authors have competing interests.

Albania to 25% in Turkey [4]. Occupational burnout can occur in any industry, such as among doctors [5], nurses [6], teachers [7], and police officers [8] Statistics from China indicate that the incidence of occupational burnout among healthcare workers is 58.0%, manufacturing employees 53.4%, delivery personnel 66.2%, chemical fiber plant workers 53.9%, and petrochemical enterprise employees 36.4% [9].

Occupational burnout has significant negative impacts on individuals and organizations across various industries. Firstly, it leads to a decrease in organizational commitment. A study on hotel industry employees found that occupational burnout is associated with reduced organizational loyalty, which subsequently results in high turnover rates and decreased job satisfaction [10]. Additionally, burnout negatively affects workplace relationships, potentially leading to conflicts that further exacerbate stress and dissatisfaction [11]. Secondly, the impact on work performance is notable. Empirical research in healthcare and education has shown that employees who report higher levels of burnout experience a decline in work efficiency. Simultaneously, absenteeism increases, ultimately affecting organizational efficiency and service quality [12,13]. Thirdly, burnout enhances turnover intention. Employees experiencing high levels of occupational burnout are more likely to consider leaving their jobs [14]. Research indicates that as burnout intensifies, employees' likelihood of seeking new employment correspondingly increases, adversely impacting workforce stability [15]. Furthermore, longitudinal studies on burnout have shown that emotional exhaustion, a core component of burnout, directly leads to employee resignation, underscoring the urgent need for organizations to develop effective strategies to address burnout [15]. Fourthly, burnout extends its effects into the realm of physical and mental health. Chronic burnout can lead to serious health issues, including anxiety, depression, and even cardiovascular diseases [16]. The chronic nature of burnout diminishes the quality of life for affected employees, raising significant ethical concerns for leadership regarding employee welfare and retention.

In the research concerning the mechanisms underlying burnout, one of the most influential contributions is the Job Demands-Resources (JD-R) model proposed by [17]. This model is grounded in the premise that job demands—defined as aspects of the job that require sustained physical or mental effort—can deplete employees' energy, leading to stress and burnout, whereas job resources—elements that help achieve work goals, foster personal growth, and mitigate job demands—can buffer against stress or motivate employees [18]. This fundamental dichotomy between positive and negative factors has fostered a dual-process view of job-related outcomes, where the interplay between job demands and resources leads to different experiences of burnout and work engagement among employees. Studies have shown that adverse working conditions, such as workload and time pressure, exacerbate burnout, while simultaneously present job resources, like social support and autonomy, can mitigate these negative effects [19,20]. The applicability of this model has been validated in the Chinese occupational context, providing reliable predictors for psychological distress and occupational burnout among employees in the non-profit sector [21], interpreting the stress process experienced by blue-collar workers [22], and decoding workplace characteristics affecting healthcare staff [23]. In its evolving

theoretical framework, scholars have added new variables to the JD-R model, increasingly incorporating the role of personal resources into its adaptive adjustment, thereby enhancing the understanding of the interaction between individual traits and the workplace [24]. Further research has found that personal traits can enhance resilience against workplace discrimination, showcasing the model's evolving nature in addressing various workplace challenges and their impacts on well-being [25].

Despite extensive exploration of burnout mechanisms in the existing literature, much of the focus has been on employees (EM) within corporate and institutional settings, with insufficient attention given to self-employed individuals such as freelancers, small business owners, and gig workers. As a critical component of the modern labor market, self-employed (SE) individuals exhibit unique work patterns characterized by high levels of autonomy and blurred boundaries of responsibility, potentially engendering distinct burnout risks. SE offers immense flexibility and control over decision-making, allowing individuals to tailor their work environments according to personal preferences and motivations [26]. However, this autonomy can also lead to prolonged stressors that exacerbate occupational burnout. Research indicates that while self-employed individuals may initially feel empowered, the pressure of managing all aspects of their business can lead to increased emotional exhaustion and feelings of being overwhelmed [27]. Additionally, the lack of resources, guidance, and community support can contribute to a sense of isolation among freelancers [28]. Consequently, these individuals, juggling marketing, client relationships, and administrative duties, may experience heightened stress, further blurring the lines between work and personal life [17]. This continuous intertwining of responsibilities without clear demarcation can lead to greater emotional fatigue and burnout, as freelancers often lack the typical "off" periods found in traditional workplace settings [16].

However, existing research has yet to systematically compare the differences in burnout between EM and SE or quantitatively reveal similarities and differences in the weighting and predictive factors of burnout dimensions across these two groups. Against this backdrop, this paper focuses on exploring three research questions: (1) Do EM and SE exhibit different manifestations of occupational burnout? (2) What are the contributing factors leading to occupational burnout among different types of workers (EM vs SE)? (3) How do the weights of these factors differ between the two groups?

The original contributions of this study mainly lie in three aspects: (1) Theoretical contribution: extending burnout theory to encompass the SE population and enriching the application scenarios of the Job Demands-Resources (JD-R) model; (2) Methodological contribution: introducing the Entropy Weight Method and Boruta Algorithm to provide a new paradigm for quantitative analysis of burnout differences; (3) Practical contribution: furnishing empirical evidence for designing differentiated mental health intervention strategies tailored to various types of employment.

The remaining sections are structured as follows: Chapter Two, Materials and Methods: This section elucidates the data sources, preprocessing methods, and quantitative research methodologies employed in this study. Chapter Three, Results: Here, we present the detailed findings of our quantitative analysis, discuss these results, compare them with relevant studies, and explore potential explanations for our observations. Chapter Four, Conclusion: In this final chapter, we summarize the key findings, acknowledge the limitations of our study, and offer perspectives for future research.

## 2 Materials and methods

### 2.1 Data sources and preprocessing

**2.1.1 Data sources and control variables.** The China Labor-force Dynamics Survey (CLDS) meticulously tracks urban and rural households and individual laborers in China, systematically monitoring changes in social structure and the interplay between families and laborers. It covers multiple dimensions such as education, employment, migration, and health, establishing a comprehensive interdisciplinary data platform. Although no new data has been added since 2018, which somewhat limits our ability to capture the latest trends, considering that mechanisms of job burnout are typically driven by long-term accumulative factors rather than short-term frequent changes, using the 2018 CLDS data for this

study remains highly relevant and valid. The strengths of the CLDS lie in its broad coverage, scientific sampling design, and detailed data content, enabling researchers to deeply investigate the work experiences and psychological states of laborers under different occupational statuses. Especially for exploring the distinct burnout mechanisms between self-employed and employed individuals, the CLDS provides rich variable resources including, but not limited to, personal backgrounds, career histories, and mental health statuses of laborers, offering irreplaceable data support for this research topic. Therefore, utilizing data from the CLDS not only helps reveal differences in job burnout between self-employed and employed individuals but also provides empirical evidence for developing targeted interventions. At the same time, we acknowledge the limitations associated with using earlier data and suggest that future research could further verify the universality and timeliness of our findings when newer data resources become available. Data is publicly stored at https://www.kaggle.com/datasets/jojoyin/clds-2018.

Table 1 presents demographic indicators of two groups (EM and SE) used as control variables, including gender, age (calculated by subtracting the birth year from the survey year 2018), educational attainment, geographical location (classified by economic development level), and their corresponding mapped values (in the Mapping column).

**2.1.2 Dependent variable.** The dependent variable employed in this study (referred to as the dependent variable in the econometric analysis section and as the target in the machine learning section) is a composite concept. To accurately measure this abstract construct, it is often necessary to rely on widely recognized scale tools, aggregating multiple observed indicators or employing factor analysis to build a "burnout index" with high reliability and validity. The introduction of the Maslach Burnout Inventory (MBI) marked a significant stage in the design of burnout scales, being the first to divide job burnout into three interrelated dimensions: emotional exhaustion, depersonalization, and reduced personal accomplishment [29]. Subsequently, Oldenburg's scale refined the measurement to focus on the first two core dimensions

**Table 1. Control variables.**

| Variables | Attribute | EM | SE | Mapping |
|---|---|---|---|---|
| gender | male | 2262 | 506 | 1 |
| | female | 1971 | 389 | 2 |
| age | mean | 40.95 | 44.86 | |
| | std | 11.76 | 10.97 | |
| | min | 15 | 17 | |
| | max | 79 | 83 | |
| education | Doctorate (PhD) | 10 | 0 | 11 |
| | Master's degree | 63 | 1 | 10 |
| | Bachelor's degree | 699 | 23 | 9 |
| | Junior college diploma | 647 | 44 | 8 |
| | Regular high school | 531 | 131 | 7 |
| | Vocational high school | 78 | 11 | 6 |
| | Secondary specialized school | 250 | 52 | 5 |
| | Technical school | 62 | 6 | 4 |
| | Junior high school | 1313 | 416 | 3 |
| | Elementary school/ Private tutor | 483 | 177 | 2 |
| | No formal education | 89 | 33 | 1 |
| location | Super first-tier cities | 203 | 6 | 5 |
| | Municipality directly under the Central Government | 106 | 20 | 4 |
| | Eastern coastal developed areas | 2271 | 436 | 3 |
| | Central relatively developed areas | 828 | 201 | 2 |
| | Western developing areas | 879 | 232 | 1 |

[30]; Pines and Aronson attempted to retain only the exhaustion dimension as the most critical aspect [31]; moreover, new simple scales applicable across various cultural or occupational backgrounds have also been developed [32].

In this study, given the limitations of available data, we opted for a set of questions from Part VII, Section Five of the questionnaire titled "Please indicate the frequency at which you experience the following states based on your feelings and experiences," as our measure of occupational burnout. The response options range from "daily/several times a week/several times a month/once a year or less/never," with corresponding mapped scores for burnout levels ranging from 5 (highest) to 1 (lowest). Four items, their corresponding variable names, MBI dimensions, and rationale are listed in Table 2.

An essential explanation of the logical mapping between the survey items and the MBI constructs is as follows: "I feel physically and mentally exhausted" directly describes Exhaustion, and Pressure is also a well-documented precursor to exhaustion [33,34,35]; "I am not interested in this" reflects a lack of personal identification with work, aligning with Depersonalization ("This doesn't interest me, so it's not my concern"); "I don't feel I accomplish meaningful work" corresponds to reduced Personal Accomplishment.

**2.1.3 Explanatory variables.** Drawing on the Job Demands-Resources (JD-R) model [17] and suggestions from its recently evolved versions [36,37,38,39], this study incorporates three constructs as explanatory variables (referred to as independent variables in subsequent econometric analyses and as features in the machine learning section). These include job demands (JD, with a positive influence), job resources (JR, with a negative influence), and personal resources. Although [40] mentioned the role of individual capital within the JD-R model, they did not further decompose it into more granular dimensions.

Innovatively, this study disaggregates the concept of personal resources (individual capital, IC) into social capital (SC) and health capital (HC). Social capital is further divided into close relationship-based family support (FS) and more distant relationship-based social support (SS); health capital is subdivided into mental health (MH) and physical health (PH). The theoretical model constructed for this study is illustrated in Fig 1.

In the process of identifying lower-order indicators/items that match higher-order constructs (JD, JR, IC), we adhered to the principle of "selecting items from the existing items in the questionnaire as much as possible, which can reasonably correspond to the target constructs in terms of theoretical logic and measurement connotation." Our judgment and selection of variables were based on this understanding: The essence of JR lies in empowering, supporting, buffering, and promoting development; the essence of JD is pressuring, depleting, restricting, and increasing burdens; the essence of FS is to provide emotional, economic, and educational resources, thereby fostering individual development. Additionally, the essence of SS is assistance and understanding from social networks, along with buffering stress; MH reflects the state

**Table 2. Burnout measurement items.**

| Item | Variable Name | MBI Construct | Rationale |
|---|---|---|---|
| Work makes me feel mentally and physically exhausted | Exhaustion | Emotional Exhaustion | Directly reflects the continuous psychological and physical burden an individual feels from work, which is a key component of emotional exhaustion. |
| Working all day is indeed stressful for me | Pressure | | Expresses the state of emotional and physical fatigue due to prolonged work stress, which is a core manifestation of emotional exhaustion. Note: This item does not directly correspond to a specific MBI construct but relates closely to the concept of emotional exhaustion. |
| I am becoming less interested in this job | Detachment | Depersonalization | Indicates a lack of enthusiasm, indifference, or distancing attitude towards the job, representing a typical psychological response in the dimension of depersonalization. |
| I feel I have accomplished many valuable tasks | Reduced Accomplishment | Reduced Personal Accomplishment | Originally meant to reflect high personal accomplishment; however, when inversely processed, it indicates "lack of achievement," suggesting one's work lacks value. It aligns with the measurement logic of decreased personal accomplishment. |

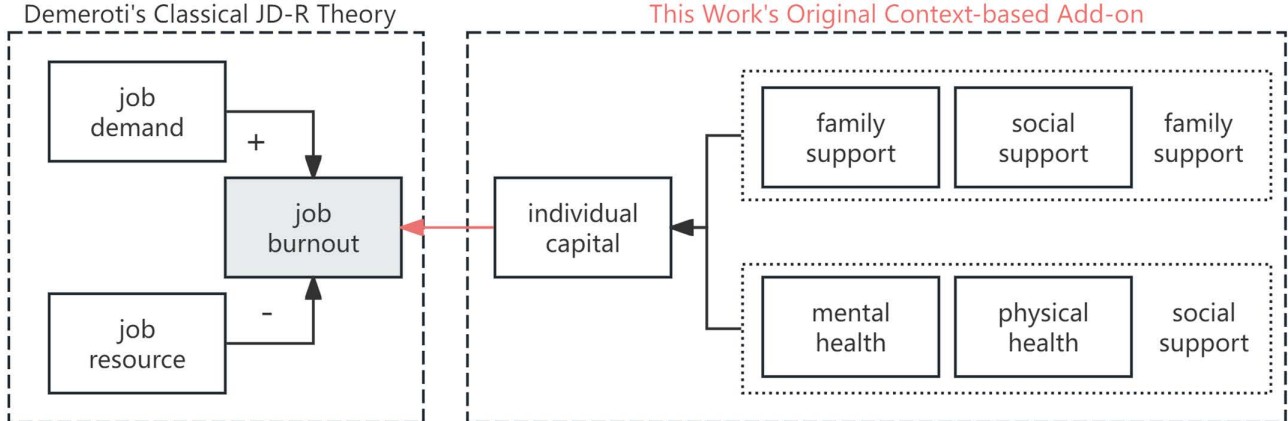

**Fig 1. Theoretical model of this study.**

of mental health and the risk of mental illness; PH encompasses behaviors and environmental factors that directly impact physical function and disease risk. In practice, due to limitations related to data availability and the way items are phrased, some logical explanations are required to fully justify the suitability of certain lower-order indicators for their corresponding higher-order constructs.

Given that questions asked in the questionnaire vary by occupational type, Tables 3 and 4 present the JD-R variables and reasons for inclusion for EM and SE respectively. Table 5 presents the selected indicators and rationale for IC, which are applicable to both EM and SE. Original items for all selected variables are provided in the S1 Appendix for reference.

Table 6 presents the multicollinearity test results among the independent variables calculated using the Entropy Weight Method (see 2.2.1). Typically, a VIF < 5 indicates acceptable levels of multicollinearity with no serious collinearity issues. The VIF values for both groups of variables are less than 2, indicating good independence among the variables and confirming their suitability for subsequent statistical analyses.

### 2.2 Analytical methods

**2.2.1 Factor weight analysis based on entropy weight method.** When synthesizing low-order indicators into high-order indicators, the entropy weight method is utilized to determine the relative weights of each indicator in the comprehensive evaluation. The final index is generated through a weighted summation approach. Initially, raw data are standardized to eliminate the influence of different dimensions and magnitudes, ensuring all indicators are converted into unitless standardized values. Subsequently, the entropy weight method is applied to calculate the information entropy of each low-order indicator, thereby measuring its information dispersion. The greater the dispersion, the more effective information the indicator contains, and thus, it should be assigned a higher weight in the comprehensive evaluation.

The specific steps are as follows: Let the total number of samples be $n$, and the number of low-order indicators be $m$. Construct a weight matrix for the $i$-th sample on the $j$-th indicator, where $y_{ij} = x_{ij}/sum_{i=1}^{n} x_{ij}$. Then calculate the information entropy $e_j = -(1/lnn) * sum_{i=1}^{n}(y_{ij} * lny_{ij})$. If $y_{ij} = 0$, then set the corresponding $y_{ij} * lny_{ij}$ value to 0 to avoid undefined situations. On this basis, further calculate the difference coefficient $y_{ij} = 0$, and normalize it to obtain the weight of each indicator $w_j = d_j/sum_{j=1}^{m} d_j$. Finally, multiply the standardized values of each sample on various low-order indicators by their corresponding weights, and then sum them up with weights to form the comprehensive index of the high-order concept, i.e., $Index_i = sum_{j=1}^{m}(w_j * z_{ij})$, where $z_{ij}$ represents the standardized score of the $i$-th sample on the $j$-th low-order

**Table 3. EM JD-R variables.**

| High-order Construct | low-order Construct | Variable Name | Rationale |
|---|---|---|---|
| Job resource | Union Assistance | union_help | Unions provide support to workers in labor rights protection, negotiation, and legal aid, serving as a typical form of organizational-level social support resources. |
| | Payment Structure | payment_level | A clear and stable salary structure enhances workers' expectations and control over income, acting as an economic incentive resource. |
| | Type of Employment Contract | position_security | The stability of a contract impacts job security; long-term or open-ended contracts can be seen as institutional protective resources. |
| | Autonomy Over Job Content | content_control | Control over job content boosts individual self-efficacy and task management abilities, constituting a crucial psychological resource. |
| | Autonomy Over Work Schedule | schedule_control | The ability to arrange work pace independently helps regulate work pressure and reduce time conflicts, enhancing adaptability. |
| | Autonomy Over Work Intensity | intensity_control | Being able to adjust work intensity indicates an individual's capacity for load management, contributing to maintaining physical and mental health as an operational resource. |
| | Position Level | position_level | Rank not only represents status but also reflects decision-making participation and resource acquisition capability, serving as an essential structural resource for career development. |
| Job demand | Tolerance for Wage Arrears | wage_delay | Accepting delayed wages means workers must endure economic uncertainty, representing involuntary economic burdens. |
| | Tolerance for Mandatory Overtime | extra_work | Being forced to extend working hours increases physical and psychological burdens, typifying temporal job demands. |
| | Tolerance for Workplace Injury | injury | Working in environments with injury risks entails individuals bearing the uncontrollable cost of bodily harm, falling under safety requirements. |
| | Tolerance for Danger | unsafety | Operating in hazardous conditions reflects an individual's tolerance for potential threats, serving as an environmental stressor. |
| | Tolerance for Pollution | pollution | Exposure to harmful environments can cause long-term damage to health, imposing physiological demands due to external environmental factors. |
| | Tolerance for Heavy Physical Labor | physical_labor | Engaging in intensive physical labor requires continuous physical exertion, leading to significant physical exhaustion, thus constituting a physical dimension of job demands. |
| | Tolerance for Frequent Physical Movement | physical_move | Frequent physical activity increases motion loads, easily causing muscle fatigue and distraction, categorizing it under motor physical requirements. |
| | Requirement for High-intensity Mental Labor | mental_labor | Prolonged high concentration on information processing exhausts cognitive resources, making it a typical cognitive job demand. |
| | Requirement for Internet Skills | internet_skill | New technical requirements in the digital age reflect mandatory expectations for employees' knowledge updating capabilities. |
| | Requirement for Specialized Training | need_training | Specific skill requirements for positions imply that employees must invest time and effort into learning, creating entry barriers. |
| | Duration of Skill Acquisition | skill_demand | The longer the learning period required to master necessary skills, the higher the preparatory costs before employment, representing implicit human capital demands. |
| | Minimum Educational Qualification | min_education | Educational thresholds limit employment opportunities, reflecting societal screening standards based on knowledge levels, acting as an educational background entrance requirement. |
| | Minimum Work Experience | min_experience | Experience thresholds increase the difficulty of meeting job requirements, necessitating workers to accumulate a certain number of years of practical experience, forming an experiential prerequisite. |

indicator. This index reflects the overall level of an individual under the high-order variable, with higher values indicating better performance in that concept. This method not only retains the integrity of the original data's information but also automatically adjusts the contribution degree of each indicator based on the data distribution, enhancing the objectivity and explanatory power of the synthesized index.

**Table 4. SE JD-R variables.**

| Higher-order Construct | Lower-order Indicator | Variable Name | Rationale |
|---|---|---|---|
| Job Resource | Entrepreneurial motivation | entre_reason | Reflects the underlying drive or reason for starting a business, indicating available internal motivation as a resource. |
| | Entrepreneurial opportunity channel | entre_channel | Refers to how entrepreneurs access opportunities, reflecting the availability of external resources. |
| | Stable relationship with government clients | gov_customers | A stable connection with government clients represents a reliable business resource. |
| | Stable relationship with institutional clients | institution_clients | Indicates long-term institutional support, functioning as a key business resource. |
| | Stable relationship with corporate clients | corporate_clients | Stable ties with corporate clients provide consistent business opportunities and thus represent a valuable resource. |
| | Stable relationship with social organizations | social_organization | Partnerships with social organizations can offer support, legitimacy, and networking, serving as a non-market resource. |
| | Stable relationship with individual customers | individual_customers | Regular customer relationships contribute to revenue stability and are considered a critical business asset. |
| | Quality channels in early entrepreneurship | busi_channel | Having good channels at the start-up phase reflects initial access to resources that support business development. |
| | Number of people who provided business help | provided_business | Shows the extent of support received from others, indicating social capital as a resource. |
| | Degree of familiarity with helpers | provided_business_known | Familiarity with supporters enhances trust and efficiency in leveraging resources. |
| | Number of people who introduced business | provided_business_num | Reflects the volume of business leads derived from personal or professional networks. |
| | High-quality channels among referrers | provider_department | Indicates the quality of business-referral sources, which is crucial for sustainable growth. |
| | Effectiveness of interpersonal relationships | relation_nohelp | Measures how well relationships facilitate business outcomes, representing relational resource effectiveness. |
| | Importance of interpersonal relationships | relation_nodeci | Reflects the perceived value of personal networks in supporting entrepreneurial activities. |
| | Number of key relationships | relation_onlykey | Highlights the quantity of critical connections, which serve as strategic resources. |
| Job Demand | Average number of rest days per month | month_rest_days | Fewer rest days indicate higher workload and time pressure, reflecting job demand. |
| | Average daily working hours | day_work_hours | Long working hours are a direct indicator of high job demands. |
| | Demand for specialized training | need_training | Requirement for specific skills indicates cognitive or technical job demands. |
| | Demand to cope with fierce competition (early stage) | competition_start | Facing strong competition in the early stages increases stress and effort required. |
| | Demand to obtain government support | gov_support | The necessity to seek policy or financial support adds administrative and procedural burden. |
| | Demand to cope with fierce competition (past year) | competition_year | Ongoing exposure to intense competition reflects sustained high job demands. |
| | Demand to develop technological barriers | skill_demand | Building technical advantages requires continuous effort and innovation. |
| | Demand for economic foundation | eco_support | Financial requirements add pressure and complexity to entrepreneurial tasks. |
| | Demand to build a network of contacts | relation_support | Establishing and maintaining relationships is labor-intensive and emotionally demanding. |
| | Demand for business experience | exp_support | Lack of experience increases the learning curve and overall job demands. |

**Table 5. IC variables.**

| Higher-order Construct | Lower-order Indicator | Variable Name | Rationale |
|---|---|---|---|
| Family Support | Father Alive | father_alive | The presence of a father impacts the completeness of family structure and source of emotional support, serving as one of the foundational conditions for family stability. |
| | Mother Alive | mother_alive | The health condition and presence of a mother directly influence the family's emotional atmosphere and care, being a core element of familial support. |
| | Paternal Education | paternal_edu | Educational attainment reflects the father's knowledge base and cognitive abilities, generally positively correlated with educational investment and support quality within the family. |
| | Maternal Education | maternal_edu | The mother's level of education often influences parenting styles, quality of parent-child communication, and the developmental environment for children. |
| | Paternal Job Type | paternal_job | The type of employer reflects the father's occupational status and social resource acquisition capability, indirectly affecting the family's social capital. |
| | Maternal Job Type | maternal_job | The nature of the mother's work also embodies her social role and resource control ability, contributing to enhancing overall family support. |
| | Socioeconomic Status at Age 14 | soeco_status | Reflects on the economic and social standing of the family when the individual was 14 years old, indicating the family resource environment during early development stages. |
| | Spouse's Financial Contribution in First Marriage | spouse_cost | Economic input from a spouse demonstrates their commitment to the marriage, representing a form of resource support within marital relationships. |
| | Parental Financial Support in Marriage | parent_cost | Financial support provided by elders for children's marriages illustrates intergenerational emotional bonds and resource transmission. |
| Social Support | Number of People Helping Find Jobs | job_hunt | The number of people assisting in job searches indicates the practicality of social capital and interpersonal networks. |
| | Number of People to Confide In | tell_truth | The ability to confide inner thoughts to others reflects whether an individual has a trustworthy emotional support system. |
| | Number of People for Discussing Important Issues | disc_imp | Being able to discuss key life issues with others signifies cognitive and decision-making support within a social network. |
| | Number of People Providing Financial Aid | borrow_money | Receiving aid during financial hardships is the most direct form of instrumental support within social support. |
| | Familiarity with Neighbors | familiarity | The degree of familiarity with neighbors reflects the frequency of social interactions and trust basis at the community level. |
| | Trust in Neighbors | mutual_trust | Evaluations of trust towards neighbors illustrate the trust network among non-kin relationships within the community. |
| | Mutual Assistance Among Neighbors | mutual_help | Actual instances of mutual assistance between neighbors represent concrete expressions of community support. |
| Mental Health | Easily Annoyed by Minor Issues | annoyed_trivial | Frequent anger and mood swings indicate emotional instability, suggesting diminished psychological adjustment functions. |
| | Poor Appetite | poor_appetite | Psychological problems often accompany physiological reactions; changes in appetite are typical symptoms of depression. |
| | Inner Distress | inner_distress | Expressions of internal pain and suppression reflect the extent of emotional distress. |
| | Low Self-esteem | low_self_esteem | A lower self-evaluation is a significant psychological characteristic indicative of compromised mental health. |
| | Concentration Difficulties | concentration_issue | Difficulty concentrating is a manifestation of affected cognitive functions, commonly seen in anxious and depressed individuals. |
| | Low Mood | low_mood | Directly reflects the core experience of depressive emotions. |
| | Effortful Execution of Tasks | task_effort | Lack of motivation and sluggish actions are typical behavioral manifestations of depression. |

**Table 5.** (Continued)

| Higher-order Construct | Lower-order Indicator | Variable Name | Rationale |
|---|---|---|---|
| | Hopelessness about the Future | hopeless_future | Loss of confidence in the future is a typical negative cognitive pattern, signaling severe psychological risks. |
| | Sense of Life Failure | life_failure | Feeling unsuccessful in life reflects lowered self-worth, indicating deterioration in mental health. |
| | Fearfulness | fearful | Persistent feelings of fear are a primary feature of anxiety disorders. |
| | Poor Sleep Quality | poor_sleep | Insomnia or poor-quality sleep is a common symptom across various psychological disorders. |
| | Unhappiness | unhappy_feeling | Absence of joy is a core marker of mood disorders. |
| | Reduced Speech | speaks_little | Social withdrawal and reduced speech are external behavioral indicators of depression. |
| | Loneliness | feels_lonely | Loneliness is a result of social isolation and a signal of deteriorating mental health. |
| | Perceived Hostility from Others | people_unfriendly | Cognitive biases towards others' hostility are common in those with psychological disturbances. |
| | Lack of Purpose in Life | life_meaningless | Absence of life meaning is a severe sign of psychological crisis. |
| | Crying Episodes | has_cried | Involuntary crying is an external manifestation of intense emotional stress. |
| | Sorrowfulness | sorrowful | Continuous sorrow is a subjective description of a depressive state. |
| | Feeling Disliked by Others | unliked_by_others | Feelings of rejection are significant triggers for low self-esteem and social anxiety. |
| | Thoughts of Inability to Continue Living | life_cant_continue | Thoughts of despair are warning signs of extreme psychological crises. |
| Physical Health | Smoking Habit | smoke_self | Smoking is recognized as one of the main risk factors for chronic diseases and premature death, directly impacting respiratory and cardiovascular health. |
| | Secondhand Smoke Exposure at Work | smoke_cowork | Long-term exposure to secondhand smoke increases cancer and respiratory disease risks, constituting a passive health threat. |
| | Drinking Habit | drink | Excessive drinking is closely linked to liver disease, nervous system damage, and other health issues, serving as an important indicator of health behavior. |

**Table 6. Results of the multicollinearity tests.**

| Variable | VIF of SE | VIF of EM |
|---|---|---|
| job_resource | 1.046 | 1.327 |
| job_demand | 1.061 | 1.344 |
| family_support | 1.001 | 1.026 |
| social_support | 1.011 | 1.014 |
| mental_health | 1.003 | 1.005 |
| physic_health | 1.014 | 1.008 |

Subsequently, since the IC variables have consistent question items for both groups of people, to evaluate whether the differences between EM and SE values in each row are significant, a Z-score method based on the difference distribution is used for judgment. Specifically, first calculate the mean $\mu_d$ and standard deviation $\sigma_d$ of the differences between employee values and entrepreneur values for all samples. Then, for each row, separately calculate the extent to which

its difference deviates from the overall mean difference, using the formula $z = \frac{|x-y-\mu_d|}{\sigma_d}$, where $x$ and $y$ represent the employee and entrepreneur values corresponding to that row, respectively. If the $z$-value of a certain row is greater than 2, it is determined that the difference in that row exceeds the typical range of the overall data differences, and it is judged as "significant difference." This method considers the distribution characteristics of the overall data and can effectively identify rows that show abnormal deviations at the individual question level.

**2.2.2 Multiple linear regression.** We initially constructed a linear regression model with job burnout as the dependent variable. The independent variables included core explanatory variables: job resources, job demands, family support, social support, mental health, and physical health; simultaneously, gender, age, education level, and place of residence were incorporated into the analysis as control variables. In the second model, robust standard errors were introduced on the basis of the same independent and control variables to enhance the robustness of the estimates. The basic form of this regression model is:

$$\text{Burnout} = \beta_0 + \beta_1 \cdot JR + \beta_2 \cdot JD + \beta_3 \cdot FS + \beta_4 \cdot SS + \beta_5 \cdot MH + \beta_6 \cdot PH + \gamma \cdot \text{control} + \varepsilon$$

Here, $\beta_0$ represents the intercept term, $\beta_1$ to $\beta_6$ represent the coefficients for each corresponding variable, $\gamma$ represents the parameter set for the control variables, $\varepsilon$ denotes the error term. By comparing these two models, it is possible to evaluate the direction and significance level of the impact of each predictor on job burnout, and to test the robustness of the results under different standard error specifications.

**2.2.3 Feature weight analysis based on Boruta.** The Boruta algorithm is a feature selection method based on random forests, suitable for variable screening in high-dimensional datasets. It also boasts advantages in terms of stability and computational efficiency, making it particularly appropriate for research scenarios requiring the identification of all relevant variables [41]. To further investigate which of the lower-order variables (i.e., individual questionnaire items) have more significant impacts on job burnout, we employed the Boruta feature selection algorithm to conduct 100 experiments with random seeds. In each experiment, the dataset was randomly divided into a training set and a test set at a ratio of 8:2, denoted as $D_{train}^{(t)}$ for the training set and $D_{test}^{(t)}$ for the test set, where $t = 1, 2, ..., 100$. Subsequently, based on the random forest model, importance scores were calculated for all features by introducing randomly generated shadow features. The importance score $I_j^{(t)}$ for each original feature was then compared with the maximum importance value of its corresponding shadow features to determine if it is statistically significant. In each experiment, the importance values of all features were recorded. After completing the 100 experiments, the arithmetic mean of the results was taken to obtain a stable importance estimate: $\bar{I}_j = \frac{1}{T} \sum_{t=1}^{T} I_j^{(t)}$, $T = 100$. Here, $\bar{I}_j$ represents the average importance value of the $j$-th feature across the 100 experiments. This method not only identifies variables that play a key role in predicting job burnout but also effectively excludes interference caused by randomness and model instability, thereby providing a reliable basis for ranking feature importance.

Although both the Entropy Weight (EW) and Boruta appear to determine weights, their algorithmic logics are fundamentally different. EW considers only the information provided by the independent variables/features themselves, It is commonly used to synthesize independent variables (composite indicators), which are then linked to the dependent variable; whereas Boruta is a machine learning algorithm that evaluates how these independent variables/features influence the dependent variable/target. In essence, EWM follows a "multiple causes, allocated weights" approach, while Boruta adopts a "working backward from the outcome to its causes" logic. Together, Boruta and EW enable both robust feature selection and nuanced weight interpretation, revealing not only which variables matter but also how much they contribute in context-specific ways.

## 3 Results

### 3.1 Regression results

Table 6 presents the outcomes of baseline regression and robust standard error regression modeling.

## 3.2 Weights of Burnout items

Fig 2 shows the weights of various burnout manifestations calculated using the entropy weight method. Although the weights of "Exhaustion" (0.201 < 0.202, diff = 0.001) and "Value" (0.155 < 0.160, diff = 0.005) are similar across employees and self-employed individuals, significant differences emerged in "Pressure" (0.246 > 0.227, diff = 0.019) and "Interest" (0.398 < 0.411, diff = 0.013).

## 3.3 Weights of JD-R items

Table 7 reveals the weight distribution of each item in the JD-R model through the entropy weight method. Coupled with the visual representation using Excel's conditional formatting, it intuitively reflects the differences between employees and self-employed individuals in terms of key influencing factors.

## 3.4 Weights of IC items

Table 8 reveals the distribution of item weights within the IC model using the entropy weight method, combined with the visual representation via Excel's conditional formatting, providing an intuitive reflection of sensitivity differences in key variables between two groups (EM & SE) (Table 9).

## 3.5 Boruta results

Fig 3 illustrates the comparison of the top 10 most important features determined by the Boruta algorithm across different occupational classifications (EM vs SE).

# 4 Discussion

## 4.1 Discussion about regression results

There are notable differences in the mechanisms influencing occupational burnout between self-employed (SE) individuals and employees (EM), reflecting distinct pathways in job characteristics, methods of accessing resources, and coping strategies for stress between these two groups.

In the aspect of job resources, the coefficient for self-employed individuals is not significant, whereas in the employee model, it shows a highly significant negative impact. This indicates that employees are more prone to burnout due to

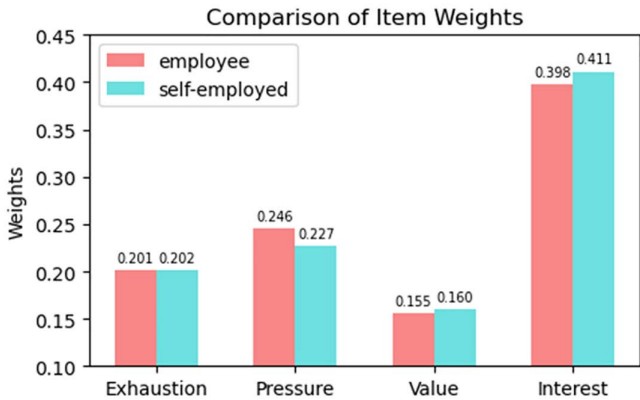

**Fig 2. Weights of burnout items.**

**Table 7. Regression result (baseline and robust).**

| | SE (base) | SE (robust) | EM (base) | EM (robust) |
|---|---|---|---|---|
| job_resource | −2.4299 | −2.4299 | −92.0175*** | −92.0175*** |
| | (−0.4641) | (−0.5350) | (−5.7485) | (−6.0510) |
| job_demand | −19.9005** | −19.9005** | 184.8383** | 184.8383** |
| | (−2.0023) | (−2.0628) | (−3.1683) | (−3.015) |
| family_suport | −0.8227 | −0.8227 | 18.4673* | 18.4673** |
| | (−0.1534) | (−0.1623) | (−1.9119) | (−2.1314) |
| social_suport | 1.0858* | 1.0858*** | −0.1408 | −0.1408 |
| | −1.8568 | −5.677 | (−0.0677) | (−0.1065) |
| mental_health | −4.2e+02*** | −4.2e+02*** | −2.3e+03*** | −2.3e+03*** |
| | (−7.8823) | (−5.7736) | (−20.7289) | (−17.6796) |
| physic_health | 50.6499 | 50.6499 | 20.3905 | 20.3905 |
| | −1.5504 | −1.5927 | −0.319 | −0.3218 |
| gender | −0.0201 | −0.0201 | −0.0171** | −0.0171** |
| | (−1.2317) | (−1.2212) | (−2.4500) | (−2.5043) |
| age | −0.0025*** | −0.0025*** | −0.0015*** | −0.0015*** |
| | (−3.5485) | (−3.6941) | (−5.3074) | (−5.2354) |
| education | −0.0202** | −0.0202*** | −0.0120*** | −0.0120*** |
| | (−3.2776) | (−3.4748) | (−5.3496) | (−5.1813) |
| location | −0.0036 | −0.0036 | −0.0055* | −0.0055* |
| | (−0.4474) | (−0.4561) | (−1.7910) | (−1.7785) |
| _cons | 1.0315*** | 1.0315*** | 1.0204*** | 1.0204*** |
| | −12.5593 | −11.1359 | −28.6168 | −26.0305 |

resource scarcity, while self-employed individuals can relatively compensate for resource inadequacies through autonomy and flexibility, and may even proactively create conditions to adapt to their work environment. This suggests that resources represent an externally dependent safeguard for employees, but for the self-employed, they are the outcome of self-construction.

Regarding job demands, SE exhibit a significant negative relationship, meaning that the more challenging the job tasks, the lower the level of burnout; conversely, employees show a significant positive relationship ($p < 0.05$), indicating that high-intensity job demands exacerbate their burnout. This finding confirms different understandings of the "stress-motivation" mechanism under two forms of labor: self-employed individuals tend to view challenges as avenues for value realization, whereas employees are more likely to perceive them as burdens. These results align with [42], further revealing notable differences in emotional responses between self-employed individuals and organizational employees when facing job demands. Such differences might stem from divergent perceptions of the significance of work challenges: self-employed individuals tend to see them as paths to self-actualization, while employees often regard them as sources of stress.

Concerning family support, data indicate that this variable is not significant in the SE model, whereas in the employee model, it demonstrates a significant positive impact. Considering that the items primarily measure economically family support (rather than mentally), these findings suggest that employees might experience additional responsibility pressure from family economic aid, thus intensifying burnout. Although self-employed individuals also need family support, its mechanism is more complex and does not show statistical significance in the short term. The Chinese idiom 'Wang Zi Cheng Long'(望子成龙), literally 'hoping one's child becomes a dragon,' vividly describes how high family expectations often translate into implicit psychological burdens. In a professional context, this phenomenon could be analogized

**Table 8. Weight of JD-R items.**

| Categoty | Variable Name | Weight |
|---|---|---|
| EM-JR | union_help | 0.48438 |
| | payment_level | 0.064679 |
| | position_security | 0.264104 |
| | content_control | 0.031333 |
| | schedule_control | 0.031301 |
| | intensity_control | 0.031170 |
| | position_level | 0.093034 |
| EM-JD | wage_delay | 0.018671 |
| | extra_work | 0.022817 |
| | injury | 0.010598 |
| | unsafety | 0.019743 |
| | pollution | 0.019966 |
| | physical_labor | 0.114957 |
| | physical_move | 0.100432 |
| | mental_labor | 0.093442 |
| | internet_skill | 0.137390 |
| | need_training | 0.047417 |
| SE-JR | skill_demand | 0.158402 |
| | min_education | 0.160503 |
| | min_experience | 0.095661 |
| | entre_reason | 0.065088 |
| | entre_channel | 0.065350 |
| | gov_customers | 0.087094 |
| | institution_clients | 0.083620 |
| | corporate_clients | 0.074507 |
| | social_organization | 0.076719 |
| | individual_customers | 0.017822 |
| | busi_channel | 0.020947 |
| | provided_business | 0.110514 |
| | provided_business_known | 0.067859 |
| | provided_business_num | 0.139033 |
| | provider_department | 0.057530 |
| | relation_nohelp | 0.034131 |
| | relation_nodeci | 0.063419 |
| | relation_onlykey | 0.036366 |
| SE-JD | month_rest_days | 0.216447 |
| | day_work_hours | 0.16384 |
| | need_training | 0.163685 |
| | competition_start | 0.08006 |
| | gov_support | 0.078771 |
| | competition_year | 0.072064 |
| | skill_demand | 0.056687 |
| | eco_support | 0.056520 |
| | relation_support | 0.056013 |
| | exp_support | 0.055912 |

**Table 9. Weight of IC items.**

| Variable Name | EM | SE | Diff. | Z-score | Sig. |
|---|---|---|---|---|---|
| father_alive | 0.017359 | 0.017333 | 0.000026 | 0.000709 | FALSE |
| mother_alive | 0.016932 | 0.016931 | 0.000001 | 0.000026 | FALSE |
| paternal_edu | 0.054232 | 0.046743 | 0.007489 | 0.204532 | FALSE |
| maternal_edu | 0.049211 | 0.0382 | 0.011011 | 0.300722 | FALSE |
| paternal_job | 0.075745 | 0.073088 | 0.002657 | 0.072564 | FALSE |
| maternal_job | 0.075745 | 0.073088 | 0.002657 | 0.072564 | FALSE |
| soeco_status | 0.032228 | 0.039935 | −0.007707 | 0.210489 | FALSE |
| spouse_cost | 0.318074 | 0.306042 | 0.012032 | 0.328607 | FALSE |
| parent_cost | 0.360473 | 0.388641 | −0.028168 | 0.769303 | FALSE |
| job_hunt | 0.486864 | 0.64033 | −0.153466 | 4.191341 | **TRUE** |
| tell_truth | 0.106563 | 0.128712 | −0.022149 | 0.604917 | FALSE |
| disc_imp | 0.109325 | 0.09218 | 0.017145 | 0.468249 | FALSE |
| borrow_money | 0.283668 | 0.12776 | 0.155908 | 4.258032 | **TRUE** |
| familiarity | 0.004844 | 0.003288 | 0.001556 | 0.042495 | FALSE |
| mutual_trust | 0.002958 | 0.002659 | 0.000299 | 0.008165 | FALSE |
| mutual_help | 0.005778 | 0.005070 | 0.000708 | 0.019335 | FALSE |
| annoyed_trivial | 0.081338 | 0.090061 | −0.008723 | 0.238237 | FALSE |
| poor_appetite | 0.069758 | 0.069069 | 0.000689 | 0.018816 | FALSE |
| inner_distress | 0.048791 | 0.047889 | 0.000902 | 0.024633 | FALSE |
| low_self_esteem | 0.058263 | 0.069479 | −0.011216 | 0.306324 | FALSE |
| concentration_issue | 0.051268 | 0.045216 | 0.006052 | 0.165286 | FALSE |
| low_mood | 0.055675 | 0.053937 | 0.001738 | 0.047465 | FALSE |
| task_effort | 0.054098 | 0.073019 | −0.018921 | 0.516756 | FALSE |
| hopeless_future | 0.050582 | 0.055498 | −0.004916 | 0.134263 | FALSE |
| life_failure | 0.0438 | 0.044019 | −0.000219 | 0.005983 | FALSE |
| fearful | 0.033607 | 0.032522 | 0.001085 | 0.029631 | FALSE |
| poor_sleep | 0.112023 | 0.102369 | 0.009654 | 0.263661 | FALSE |
| unhappy_feeling | 0.056189 | 0.056034 | 0.000155 | 0.004232 | FALSE |
| speaks_little | 0.049210 | 0.037991 | 0.011219 | 0.306403 | FALSE |
| feels_lonely | 0.043132 | 0.040865 | 0.002267 | 0.061913 | FALSE |
| people_unfriendly | 0.030988 | 0.030780 | 0.000208 | 0.005679 | FALSE |
| life_meaningless | 0.034313 | 0.034968 | −0.000655 | 0.01789 | FALSE |
| has_cried | 0.027985 | 0.029593 | −0.001608 | 0.043918 | FALSE |
| sorrowful | 0.048936 | 0.044732 | 0.004204 | 0.114815 | FALSE |
| unliked_by_others | 0.026475 | 0.021754 | 0.004721 | 0.128935 | FALSE |
| life_cant_continue | 0.023568 | 0.020202 | 0.003366 | 0.091928 | FALSE |
| smoke_self | 0.297375 | 0.325003 | −0.027628 | 0.754555 | FALSE |
| smoke_cowork | 0.466291 | 0.43833 | 0.027961 | 0.763647 | FALSE |
| drink | 0.236334 | 0.236666 | −0.000332 | 0.009069 | FALSE |

as follows: when families provide material support to intervene in an individual's professional life, they convey not only resources but also responsibilities and expectations. For employees, such external support may increase anxiety about job stability and performance, forming a "giving-returning" pressure chain. For self-employed individuals, family support might be a crucial foundation for starting a business, but given the long-term and uncertain nature of their career paths,

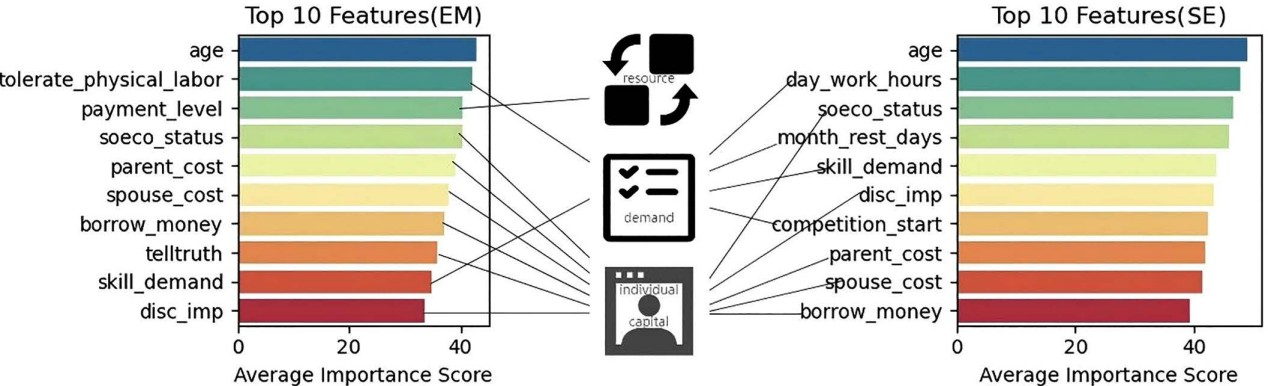

**Fig 3. Comparison of the top 10 most important features.**

there is no clear evidence of significant burnout effects in the short term. This also suggests that understanding the relationship between social support and occupational burnout should go beyond a simplistic linear thinking of "more support is better," focusing instead on underlying cultural backgrounds, group characteristics, and psychological mechanisms. In China's social structure, which emphasizes family ties and collective responsibility, family support serves both as a resource and potentially as a "gentle pressure." Future research could further integrate qualitative interviews to delve into how different types of workers perceive and cope with family support and expectations, thereby constructing a more comprehensive framework for interventions against occupational burnout.

In terms of social support, the variable shows a significant positive impact in the self-employed model, indicating that more social resources are associated with higher levels of burnout. Conversely, this variable is not significant in the employee model. This counterintuitive finding may reflect the sensitivity of self-employed individuals to "social redundancy"—excessive social support might dilute the challenges and autonomy of work, leading to burnout. In contrast, employees do not exhibit an urgent need for social resources, with their burnout being more influenced by internal organizational factors.

Mental health has a highly significant negative impact on occupational burnout for both groups, suggesting that poorer mental or physical states increase the risk of burnout. However, due to the indirect measurement of physical health through limited indicators such as active smoking, passive smoking, and drinking, although the coefficient is positive, it does not reach significance. This limitation slightly hints at unhealthy behaviors potentially serving as outlets for stress and burnout.

Gender reveals that men are more prone to burnout than women, especially among employees, but not significantly so among the self-employed. This gender tendency may be linked to occupational characteristics and societal role expectations. In traditional Chinese culture, the notion that 'Yang Jia Hu Kou' (养家糊口), literally meaning 'supporting the family and filling the mouth,' ingrains a sense of responsibility that makes male employees particularly susceptible to institutional pressures, income instability, or career stagnation. The flexible nature of self-employment and its reliance on diverse income sources may alleviate some of these pressures. Additionally, male employees may suppress emotions and neglect mental health issues to conform to societal expectations of being "strong and reliable," thereby accumulating burnout. Female employees, despite facing workplace stress, benefit from lower familial economic responsibilities, stronger emotional expression tendencies, and better utilization of social support, which may buffer against burnout.

Age and education level show significant negative impacts across both groups, with younger and less educated individuals experiencing higher burnout. This could be attributed to immature psychological adaptability in younger populations and repetitive job roles among those with lower education levels. A study on Chinese clinical nurses also found that

younger staff under 35 were more prone to emotional exhaustion due to lack of experience, but high resilience capital could mitigate this risk [43].

Finally, location was only weakly significant in the employee model, indicating that the less developed a city is, the higher the level of employee burnout, which may be related to fewer employment opportunities and restricted career development. However, there was no significant difference in burnout levels among self-employed individuals across different regions, suggesting their stronger adaptability to various locations and greater mobility.

## 4.2 Discussion about Burnout items weights

The present findings show that while "Exhaustion" and "Value" are comparably weighted between the two groups, employees report significantly higher levels of "Pressure," whereas self-employed individuals exhibit notably stronger weighting on "Interest." This pattern suggests a fundamental divergence in the core drivers of occupational burnout: employees are more susceptible to external, structural stressors, while self-employed individuals are more vulnerable to burnout stemming from diminished intrinsic motivation. Specifically, burnout among employees appears to originate primarily from institutional pressures—such as excessive workload and rigid time constraints—whereas for the self-employed, it is more closely tied to a loss of personal interest or engagement in their work. These results highlight the heterogeneity of burnout mechanisms across employment statuses and underscore the need for tailored interventions that address the distinct psychosocial contexts of each group.

The difference may be closely related to the nature of work – employees typically face clear performance evaluations and organizational norms, leading to a stronger perception of pressure. In contrast, while self-employed individuals have higher autonomy, long-term lack of interest-driven work content, such as repetitive tasks and customer management, may diminish their sense of value and commitment, reducing enthusiasm for work. From this, we can infer that employee burnout is more "externally driven," whereas self-employed individuals' tendency towards burnout is "internally driven".

The above findings resonate with some existing studies. Research has confirmed that performance pressure depletes self-regulatory resources through threat assessment, leading to negative behavior [44]. It was found that work engagement among different types of solo self-employed is mediated by intrinsic job resources like skill autonomy; repetitive tasks erode resource accumulation, providing a mechanism explanation for this study [45]. While the proposition of an "advantage-interest job crafting" model where interest enhances commitment through a chain mediation of person-job fit→calling at work→sense of meaning, also offering an explanation for the high weight of interest [46].

## 4.3 Discussion about JD-R items weights

For employees, "whether the union can provide real help" is marked with the highest weight, a phenomenon that directly reflects the core role of unions as institutional safeguards in resource acquisition. Unions mitigate job burnout caused by resource scarcity or institutional injustice through collective bargaining for better benefits, training opportunities, and rights protection for employees. Next, the sub-important (dark orange) marking of "type of labor contract" further indicates that long-term stable labor contracts often mean higher job levels and resource accessibility (such as promotion channels, performance rewards, etc.). This stability significantly reduces burnout risk through the "economic security-control sense" chain; whereas frequent uncertainties under temporary contracts exacerbate individual resource anxiety and loss of control.

In contrast, for self-employed individuals, the highest weight (red) item focuses on "number of rest days per month". The sub-important items (dark orange) include "number of rest days per week", "number of people introducing business", and "whether work requires specialized training", collectively reflecting the dual contradictions between labor intensity and resource dependence. A reduction in rest days (monthly/weekly) directly indicates high labor load for the self-employed, with this continuous physical exhaustion accelerating burnout through the "energy depletion-value collapse" pathway;

while the weight of "number of people introducing business" highlights the decisive role of market network quality in resource acquisition—richer customer resources enable self-employed individuals to alleviate economic pressure and rebuild value recognition through business expansion. However, the high weight of "whether work requires specialized training" reveals another side: the increase in skill thresholds means higher learning costs and competitive pressures. If individuals cannot quickly master high-level skills or if there is insufficient market demand for their skills, they may fall into a burnout cycle due to an imbalance between "investment-return".

The differences essentially stem from fundamental distinctions in resource acquisition models and risk-bearing mechanisms between employees and the self-employed: employees rely on institutional resources and stability provided by organizations, while the self-employed must coordinate resource acquisition and maintenance amidst autonomous decision-making and market uncertainty, thus forming two distinct patterns of occupational burnout: employees experiencing "deficiency-induced burnout" due to the absence of safeguard mechanisms, and self-employed individuals facing "imbalance-induced burnout" due to difficulties in sustaining supply-demand balance.

The above findings engage in dialogue with existing research. [47] demonstrated that resource scarcity leads to "disengagement", while adequate resources promote job engagement. [48] proposed that self-employed individuals need to dynamically adjust capabilities in response to unexpected market changes, with coordination failures leading to burnout.

### 4.4 Discussion about IC items weights

"Number of people providing help in social networks when finding a job" and "Degree of exposure to secondhand smoke at work" are assigned the highest weights across both groups. These high-weight indicators collectively point towards a latent mechanism: the imbalance in resource acquisition and structural differences in health risk exposure may indirectly affect individuals' psychological and physiological states through pathways such as "social support-stress buffering" or "environmental risk-physiological load".

Considering the inter-group differences calculated by Z-scores (EM vs. SE), significant and directional differences can be observed for two critical indicators. Specifically, "number of people providing economic support" and "number of people providing help in social networks when finding a job", both fall under the dimension of social network support. However, the weights derived from the entropy weight method essentially reflect differences in data dispersion within this dimension for the two groups. More precisely, employees may rely more on informal social networks to cope with economic pressures, such as borrowing money from friends and family or temporary subsidies; self-employed individuals tend to utilize social networks to expand business opportunities, like client referrals and industry collaborations.

Notably, these two significantly different indicators are situated within the dimension of social network support, pointing towards a phenomenon: there is a structured heterogeneity in the distribution of social network support among the two groups—employees prefer to obtain survival resources through social networks, while self-employed individuals depend on them for market resource accumulation. This difference is not driven by a single factor but likely results from the interaction of labor form (employment vs. self-employment) and resource mobilization strategies (passive reception vs. active construction).

### 4.5 Discussion about Boruta results

The analysis reveals that, among EM, only the "method of salary calculation" as a JR feature ranks within the top three most important features, indicating that for EM, the compensation structure significantly impacts occupational burnout. Further observations show that out of the top 10 important features, only two are from the JD dimension in the EM group, while there are four such features in the SE group. This suggests that occupational burnout among SE is more influenced by the nature and intensity of work tasks. Despite these features being considered highly significant, JD-R variables do not yet cover half of the top ten features.

Conversely, six of the top 10 features in the EM group and five in the SE group originate from personal capital variables. This finding supplements the classic JD-R model by emphasizing the critical role and high sensitivity of personal

capital variables in predicting the risk of occupational burnout. Specifically, whether it is enhancing social network support to improve the ability to cope with high-intensity job demands or improving physical and mental health to enhance stress resistance, personal capital demonstrates its significant role in mitigating or exacerbating occupational burnout.

### 4.6 Practical discussion beyond data limitations

The data limitation of this study is that the most recent wave from the public database CLDS used in this research stops at 2018. After the outbreak of the COVID-19 pandemic in 2019, unexpected events in the real world (rather than in theoretical study) have certainly had an impact on the overall occupational burnout in society.

The first and most direct impact is observed among healthcare workers, who are most closely associated with the COVID-19 pandemic. Multiple pieces of literature indicate that, COVID-19 pandemic and labor market shifts, including redeployment and virtual care transitions, synergistically exacerbate burnout in healthcare workers by heightening work demands while diminishing psychological resources, disproportionately affecting younger and less experienced staff [49,50,51].

The pandemic has also significantly increased burnout across non-healthcare sectors, driven by abrupt shifts to remote work, job insecurity, and intensified workloads. Employees in corporate, education, and social services faced blurred work-life boundaries and prolonged uncertainty, leading to emotional exhaustion and disengagement [52,53,54]. Furloughs and restructuring placed heavier responsibilities on remaining staff, often without adequate support, exacerbating feelings of being overwhelmed [55]. Ongoing hybrid work models and economic instability continue to pose risks, underscoring the need for organizational interventions such as resilience training and mental health support to mitigate long-term burnout in the post-pandemic labor market [56,57].

In summary, the 2019 pandemic has had widespread effects on burnout among healthcare workers (due to heightened work demands and diminished psychological resources) and non-healthcare sectors (due to remote work challenges and job insecurity), making targeted organizational interventions essential to mitigate the long-term risk of burnout in the post-pandemic era.

## 5 Conclusion

This study employs the entropy weight method, multiple linear regression, and the Boruta algorithm to delve into the differences in the manifestation and influencing factors of occupational burnout between employees (EM) and self-employed individuals (SE). Key findings indicate significant differences in resource dependence, stress coping, and paths to value realization: for EM, resource security plays a crucial role in reducing burnout; for SE, autonomy and market network quality are pivotal in mitigating burnout. Moreover, personal capital variables such as mental health, family, and social support show high sensitivity in predicting burnout risk and hold significant positions in both groups. Specifically, EM suffer from "deprivation-type burnout" due to institutional pressures and lack of external resources, whereas SE experience "imbalance-type burnout" due to difficulties balancing market demands with resource accumulation. These results not only expand the application scenarios of the classic JD-R model but also provide empirical evidence for studying occupational heterogeneity in burnout mechanisms and developing differentiated mental health intervention strategies tailored to different job types.

Despite uncovering valuable insights, this study has certain limitations. Firstly, data sourced from the CLDS 2018 may limit the capture of recent trends. Future research should utilize updated datasets to verify the timeliness and universality of current findings.

Secondly, limitations in questionnaire design led to insufficient measurement of certain variables (e.g., physical health), affecting deeper exploration of their relationship with occupational burnout. Subsequent studies could consider using more detailed measurement tools or combining qualitative methods to address these gaps. Lastly, focusing mainly on cross-sectional data, this study fails to capture changes over time; it is recommended that future longitudinal studies be

conducted to better understand the developmental trajectory and underlying mechanisms of occupational burnout. In summary, further exploration of burnout mechanisms among different types of workers not only enriches theoretical frameworks but also provides more precise support for practical interventions.

Thirdly, this study uses data from mainland China, and its generalizability awaits verification through future cross-cultural research. However, the authors believe that the JD-R model and our original extension (highlighted in red in Fig 1) constitute a universally applicable conceptual framework, which future studies can adopt to validate with data from other countries.

## Supporting information

**S1 Appendix. Original survey questions and options.**
(DOCX)

## Author contributions

**Conceptualization:** Mengjiao Yin.

**Data curation:** Mengjiao Yin.

**Formal analysis:** Mengjiao Yin.

**Funding acquisition:** Mengjiao Yin.

**Investigation:** Mengjiao Yin.

**Methodology:** Mengjiao Yin.

**Project administration:** Mengjiao Yin.

**Resources:** Mengjiao Yin.

**Software:** Mengjiao Yin.

**Supervision:** Mengjiao Yin.

**Validation:** Mengjiao Yin.

**Visualization:** Mengjiao Yin.

**Writing – original draft:** Mengjiao Yin.

**Writing – review & editing:** Mengjiao Yin, Yingying Xia.

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
