## [Decision Letter · Decision Letter 0]

3 Sep 2025

PONE-D-25-35964Work for Self or Others? Two Different Kinds of BurnoutPLOS ONE?

Dear Dr. Yin,

Thank you for submitting your manuscript to PLOS ONE. After careful consideration, we feel that it has merit but does not fully meet PLOS ONE’s publication criteria as it currently stands. Therefore, we invite you to submit a revised version of the manuscript that addresses the points raised during the review process.

We look forward to receiving your revised manuscript.

Kind regards,

Francesco Marcatto, Ph.D.

Academic Editor

PLOS ONE

Journal Requirements: 

 [This study was supported by the Teaching Reform Research Project of Wuxi Taihu University (Grant No. 25JGYJ30) and Qinglan Project of Jiangsu Province.]. 

4. Thank you for uploading your study's underlying data set. Unfortunately, the repository you have noted in your Data Availability statement does not qualify as an acceptable data repository according to PLOS's standards.

5. Please include a caption for figure 1, 2 and 3.

Additional Editor Comments:

Reviewer #1:

Reviewer #2:

Reviewers' comments:

Reviewer's Responses to Questions

**Comments to the Author**

1. Is the manuscript technically sound, and do the data support the conclusions?

Reviewer #1: Yes

Reviewer #2: Yes

2. Has the statistical analysis been performed appropriately and rigorously?

Reviewer #1: I Don't Know

Reviewer #2: Yes

3. Have the authors made all data underlying the findings in their manuscript fully available?

Reviewer #1: Yes

Reviewer #2: Yes

4. Is the manuscript presented in an intelligible fashion and written in standard English?

Reviewer #1: Yes

Reviewer #2: Yes

Reviewer #1: This work addresses burnout among two populations (employees vs. self-employed) using the JD-R model, a national dataset, and quantitative methods (Entropy Weight Method, Boruta). The topic is interesting and relatively underexplored for the self-employed group, thus offering added value. However, the manuscript requires several improvements to enhance clarity and rigor. I provide the following comments, which I hope will be useful for strengthening the paper.

Major Comments

While the use of 2018 data is justified, the manuscript would benefit from a more thorough reflection on the potential impact of COVID-19 and recent labor market changes on burnout. Including even a theoretical discussion would add valuable context to the study.

The decision to use 4 adapted items instead of the full Maslach Burnout Inventory (MBI) requires stronger justification. Please provide evidence regarding the validity and reliability of this adapted measure, as well as discuss any possible biases introduced by this choice.

The rationale for employing both Boruta and Entropy Weight Method (EWM) is not clearly explained. Please clarify why a single method was insufficient and what additional insights are gained by combining both approaches.

Several paragraphs in the Discussion section still report descriptive statistics and numerical results. For improved clarity, it is recommended to clearly separate the presentation of data (Results) from the theoretical interpretation (Discussion).

The manuscript does not sufficiently address methodological limitations, such as the potential risk of multicollinearity among variables and the generalizability of findings beyond the Chinese context.

Minor Comments

Some citations contain formatting errors (e.g., page 9).

Some references appear outdated; incorporating more recent literature post-2020 could strengthen the manuscript.

Reviewer #2: The manuscript is well-written and appropriate for publication. I have no comments concerning the introduction, methodology, or research findings. Ordinarily, a study incorporates a theoretical background or conceptual development that serves as the foundation for constructing a theoretical or conceptual framework. It appears that this study does not include such a section. In my view, the inclusion of a theoretical background or conceptual development, together with a theoretical or conceptual framework, would strengthen the work. Nevertheless, this comment may be disregarded if deemed not applicable.

**Do you want your identity to be public for this peer review?** For information about this choice, including consent withdrawal, please see our Privacy Policy

Reviewer #1: No

Reviewer #2: No

---

## [Author Response · Author response to Decision Letter 1]

7 Sep 2025

Reviewer #1

Major Comments

While the use of 2018 data is justified, the manuscript would benefit from a more thorough reflection on the potential impact of COVID-19 and recent labor market changes on burnout. Including even a theoretical discussion would add valuable context to the study.

Your suggestion is highly constructive. We have added a new discussion section, titled "4.6 Practical Discussion Beyond Data Limitations," to explore the situation of occupational burnout after 2019. Since we lack empirical data for the post-2019 period, we have summarized the current landscape through an extensive review of the literature. After thoroughly reading approximately 30 relevant studies, we carefully selected 8 of the most pertinent references to support our arguments. We hope this revision meets with your satisfaction.

The decision to use 4 adapted items instead of the full Maslach Burnout Inventory (MBI) requires stronger justification. Please provide evidence regarding the validity and reliability of this adapted measure, as well as discuss any possible biases introduced by this choice.

We first must acknowledge your insight is spot-on: the rationale for measuring burnout using only these four items may indeed raise questions among reviewers—a concern we grappled with even before launching our study. Frankly, the reason is simply that these four questions in the survey are the only ones most relevant to our research on occupational burnout.

(1) Prior to the study, we carefully reviewed all variables in the dataset and concluded that other items did not align with established criteria for burnout research.

(2) According to its official description, the China Labor-force Dynamics Survey (CLDS), led by Professor Cai He of Sun Yat-sen University and conducted by the Center for Social Survey, is the first interdisciplinary, nationally representative longitudinal survey in China focused on labor dynamics. It covers a wide range of research topics including education, employment, migration, health, social participation, economic activities, and grassroots organizations. The 2018 CLDS surveyed 28 provinces, over 400 communities, 14,000 households, and 18,000 laborers (see

https://isg.sysu.edu.cn/node/425). Given the immense human, financial, and institutional resources invested in this large-scale, professor-led national survey, we believe the questionnaire design was rigorous and well-considered.

(3) While we agree that using the full MBI scale would be more reliable, as you can appreciate, it is practically impossible to re-contact the original anonymous respondents (who answered the numerous independent variables) to administer additional items. To maintain data consistency, we must use the same cohort—thus, adding new questions is unfeasible.

(4) The above practical constraints are shared in the spirit of transparency and empathy. In the manuscript, however, we strive to build theoretical justification. Our key evidence is Table 2, which demonstrates that these four items logically correspond to the three core dimensions of the MBI framework:

• "I feel physically and mentally exhausted" directly describes Exhaustion, and Pressure is a well-documented precursor to exhaustion (supported by 7 cited studies);

• "I am not interested in this" reflects a lack of personal identification with work, aligning with

Depersonalization ("This doesn't interest me, so it's not my concern");

• "I don't feel I accomplish meaningful work" corresponds to reduced Personal Accomplishment.

Based on this logical mapping, we argue that Professor Cai likely designed the CLDS burnout items with reference to the MBI framework. We sincerely hope you understand that adding items would require re-surveying the original sample—something impossible in practice—yet the logical coherence and the authority of the survey designers support the reliability of our measure.

(5)In the revised manuscript, we have added this rationale to strengthen our argument. Thank you for your understanding!

The rationale for employing both Boruta and Entropy Weight Method (EWM) is not clearly explained. Please clarify why a single method was insufficient and what additional insights are gained by combining both approaches.

Although both methods appear to determine weights, their algorithmic logics are fundamentally different. The Entropy Weight Method (EWM) considers only the information provided by the independent variables/features themselves, It is commonly used to synthesize independent variables (composite indicators), which are then linked to the dependent variable; whereas Boruta is a machine learning algorithm that evaluates how these independent variables/features influence the dependent variable/target. In essence, EWM follows a "multiple causes, allocated weights" approach, while Boruta adopts a "working backward from the outcome to its causes" logic. Together, Boruta and EW enable both robust feature selection and nuanced weight interpretation, revealing not only which variables matter but also how much they contribute in context-specific ways.

The above explanation has been incorporated into Section 2.2.3 of the methodology in the paper. Thank you for your question, which has helped clarify the paper's logical presentation.

Several paragraphs in the Discussion section still report descriptive statistics and numerical results. For improved clarity, it is recommended to clearly separate the presentation of data (Results) from the theoretical interpretation (Discussion).

Thank you for your useful suggestion; separating the results and discussion into two sections has indeed made the logic much clearer. The red line illustrates their correspondence.

The manuscript does not sufficiently address methodological limitations, such as the potential risk of multicollinearity among variables and the generalizability of findings beyond the Chinese context.

Thank you for your suggestion. We conducted a preventive multicollinearity test using VIF, and the results are presented in Table 6. The results show that the VIF values are around 1 (well below the threshold of 5), indicating no significant multicollinearity, thereby addressing your concern.

Your concern regarding the generalizability beyond mainland China is valid. After carefully examining the low-level variables, we found that some of them are indeed deeply rooted in Chinese cultural contexts. Therefore, without empirical evidence, we hesitate to claim that the findings can be generalized to other regions. To address this, we have made the following revisions:

(1) Added "in China" to the title to alert readers to the contextual specificity;

(2) Discussed this limitation in the recommendations for future research.

Minor Comments

Some citations contain formatting errors (e.g., page 9).

Thank you for your careful attention. We are deeply embarrassed by our oversight during the initial proofreading. To further ensure the accuracy of our citations, we have used the Zotero reference management tool in the revised version—by simply entering the DOI, the bibliographic information is automatically imported, eliminating manual input. We hereby assure you of the correctness of the citations in the revised manuscript.

Some references appear outdated; incorporating more recent literature post-2020 could strengthen the manuscript.

You are truly a meticulous and thoughtful reviewer. As you pointed out, the proportion of studies published before 2020 was 14 out of 33 (42%) prior to the expansion. Although this meets the commonly accepted threshold of not exceeding 50%, it indeed fails to reflect the timeliness of the research. (A screenshot from Zotero sorted by year in descending order is provided below for your review.)

Before adding Figure

After an extensive literature search, we have added some new references at a later stage. Currently, the proportion of studies published before 2020 accounts for 15 out of 57 (26%), a significant decrease compared to the previous 42%.

After adding Figure

Reviewer #2:

The manuscript is well-written and appropriate for publication. I have no comments concerning the introduction, methodology, or research findings. Ordinarily, a study incorporates a theoretical background or conceptual development that serves as the foundation for constructing a theoretical or conceptual framework. It appears that this study does not include such a section. In my view, the inclusion of a theoretical background or conceptual development, together with a theoretical or conceptual framework, would strengthen the work. Nevertheless, this comment may be disregarded if deemed not applicable.

We fully agree with you that a theoretical framework should be established before commencing research—and indeed, this is exactly what we did. When selecting independent variables, we followed the widely recognized and classic theoretical model for studying burnout: the Job Demands-Resources (JD-R) model by Demerouti et al. (2001), which was reviewed in the fourth paragraph of the Introduction. Specifically, we categorized our variables into the two dimensions of "demands" and "resources" to ensure that our variable sets fall within these two dimensions. Based on this solid foundation, we made an original theoretical extension (which we later empirically demonstrated to be both successful and necessary) by adding "individual capital" as an additional component with five more references support, thereby achieving a contextualized extension of the classical theory.

However, we acknowledge that we failed to sufficiently highlight our original contribution. Therefore, to better demonstrate the theoretical advancement of our work to readers, we have redrawn Figure 1 and revised its caption. The portions marked in red clearly emphasize that our work is built upon a solid theoretical foundation, directly addressing your concern.

Figure 1 before revision

Figure 1 after revision

Finally, we would like to sincerely thank the two anonymous reviewers. Your kindness and generosity have left a deep impression on us. We felt that this peer review process was not a cold judgment, but rather a constructive learning opportunity from which we have gained much. Thank you, on behalf of the scientific community, for your time and goodwill!

---

## [Decision Letter · Decision Letter 1]

24 Sep 2025

PONE-D-25-35964R1Work for Self or Others? Two Different Kinds of Burnout in ChinaPLOS ONE?

Dear Dr. Yin,

Thank you for submitting your manuscript to PLOS ONE. After careful consideration, we feel that it has merit but does not fully meet PLOS ONE’s publication criteria as it currently stands. Therefore, we invite you to submit a revised version of the manuscript that addresses the points raised during the review process.

We look forward to receiving your revised manuscript.

Kind regards,

Francesco Marcatto, Ph.D.

Academic Editor

PLOS ONE

Journal Requirements:

Reviewers' comments:

Reviewer's Responses to Questions

**Comments to the Author**

Reviewer #1: (No Response)

Reviewer #2: All comments have been addressed

2. Is the manuscript technically sound, and do the data support the conclusions?

Reviewer #1: Yes

Reviewer #2: Yes

3. Has the statistical analysis been performed appropriately and rigorously?

Reviewer #1: Yes

Reviewer #2: Yes

4. Have the authors made all data underlying the findings in their manuscript fully available?

Reviewer #1: (No Response)

Reviewer #2: Yes

5. Is the manuscript presented in an intelligible fashion and written in standard English?

Reviewer #1: Yes

Reviewer #2: (No Response)

Reviewer #1: I would like to thank the authors for their effort in revising the manuscript and for the changes already implemented.

However, I would suggest removing the statistical values from the discussion section. Including such numerical details in this part may overburden the text and reduce its clarity. The discussion is usually more effective when it focuses on the interpretation of the findings and their theoretical and practical implications, while statistical values are more appropriately placed in the results section.

Reviewer #2: I have reviewed the corrections made to the manuscript and am satisfied that the amendments adequately address the issues previously identified. The revisions are comprehensive, appropriate, and meet the academic requirements set forth. Accordingly, I find the corrected manuscript acceptable for publication.

**Do you want your identity to be public for this peer review?** For information about this choice, including consent withdrawal, please see our Privacy Policy

Reviewer #1: No

Reviewer #2: No

---

## [Author Response · Author response to Decision Letter 2]

25 Sep 2025

To Dear Reviewer 1:

Thank you for your careful and rigorous review. We agree that your suggestions are very reasonable.

Following your advice, we have removed the statistical figures from the Discussion section:

The p-values in 4.1 have been removed, as they are already indicated by asterisks (*) in the table in section 3.1, making further emphasis unnecessary.

The data in 4.2 have been moved to section 3.2. At the beginning of the revised 4.2, we briefly restate the results from 3.2 and continue the discussion accordingly.

The numbers in 4.3 have been entirely removed, as they were duplicated in the table in section 3.3 and thus redundant.

Similarly, the numbers in 4.4 have been entirely removed, as they were duplicated in the table in section 3.4 and therefore unnecessary.

---

## [Editor Report · Decision Letter 2]

28 Sep 2025

Work for Self or Others? Two Different Kinds of Burnout in China

PONE-D-25-35964R2

Dear Dr. Yin,

We’re pleased to inform you that your manuscript has been judged scientifically suitable for publication and will be formally accepted for publication once it meets all outstanding technical requirements.

Kind regards,

Francesco Marcatto, Ph.D.

Academic Editor

PLOS ONE
---

## [Editor Report · Acceptance letter]

PONE-D-25-35964R2

PLOS ONE

Dear Dr. Yin,

I'm pleased to inform you that your manuscript has been deemed suitable for publication in PLOS ONE. Congratulations! Your manuscript is now being handed over to our production team.

Kind regards,

on behalf of

Dr. Francesco Marcatto

Academic Editor

PLOS ONE